# The miR-19a/Cylindromatosis Axis Regulates Pituitary Adenoma Bone Invasion by Promoting Osteoclast Differentiation

**DOI:** 10.3390/cancers16020302

**Published:** 2024-01-11

**Authors:** Zhuowei Lei, Quanji Wang, Qian Jiang, Huiyong Liu, Linpeng Xu, Honglei Kang, Feng Li, Yimin Huang, Ting Lei

**Affiliations:** 1Department of Orthopedics, Tongji Hospital of Tongji Medical College of Huazhong University of Science and Technology, Jiefang Avenue. 1095, Wuhan 430030, China; 2Sino-German Neuro-Oncology Molecular Laboratory, Department of Neurosurgery, Tongji Hospital of Tongji Medical College of Huazhong University of Science and Technology, Jiefang Avenue. 1095, Wuhan 430030, China

**Keywords:** pituitary adenoma, microRNA-19, osteoclast, cylindromatosis, tumor necrosis factor receptor-associated factors 6, ubiquitination

## Abstract

**Simple Summary:**

In this study, we demonstrate that miR-19a promotes the progression of bone invasion of pituitary adenoma and reveal the specific mechanism. miR-19a is released from pituitary adenoma cells into the tumor microenvironment and enters the osteoclasts to exert its effects. miR-19a reduces cylindromatosis (CYLD) expression, thereby preventing the K63 deubiquitination of tumor necrosis factor receptor-associated factor 6 (TRAF6). k63 ubiquitinated TRAF6 acts as a key downstream node of RANK and can activate downstream NF-кB and mitogen-activated protein kinase (MAPK) pathways to promote osteoclastogenesis. Through a combination of molecular biology and cytology, as well as in vivo experiments, we verified the specific mechanism by which miR-19a promotes bone invasion in pituitary tumors.

**Abstract:**

Background: The presence of bone invasion in aggressive pituitary adenoma (PA) was found in our previous study, suggesting that PA cells may be involved in the process of osteoclastogenesis. miR-19a (as a key member of the miR-17-92 cluster) has been reported to activate the nuclear factor-кB (NF-кB) pathway and promote inflammation, which could be involved in the process of the bone invasion of pituitary adenoma. Methods: In this work, FISH was applied to detect miR-19a distribution in tissues from patients with PA. A model of bone invasion in PA was established, GH3 cells were transfected with miR-19a mimic, and the grade of osteoclastosis was detected by HE staining. qPCR was performed to determine the expression of miR-19a throughout the course of RANKL-induced osteoclastogenesis. After transfected with a miR-19a mimic, BMMs were treated with RANKL for the indicated time, and the osteoclast marker genes were detected by qPCR and Western Blot. Pit formation and F-actin ring assay were used to evaluate the function of osteoclast. The TargetScan database and GSEA were used to find the potential downstream of miR-19a, which was verified by Co-IP, Western Blot, and EMSA. Results: Here, we found that miR-19a expression levels were significantly correlated with the bone invasion of PA, both in clinical samples and animal models. The osteoclast formation prior to bone resorption was dramatically enhanced by miR-19, which was mediated by decreased cylindromatosis (CYLD) expression, increasing the K63 ubiquitination of tumor necrosis factor receptor-associated factor 6 (TRAF6). Consequently, miR-19a promotes osteoclastogenesis by the activation of the downstream NF-кB and mitogen-activated protein kinase (MAPK) pathways. Conclusions: To summarize, the results of this study indicate that PA-derived miR-19a promotes osteoclastogenesis by inhibiting CYLD expression and enhancing the activation of the NF-кB and MAPK pathways.

## 1. Introduction

Bone homeostasis is associated with bone breakdown (osteoclastic resorption) and build-up (neo-osteoblastic formation) [1,2]. A bone remodeling imbalance causes endocrine disorders and various bone disorders, including tumor-related disease, osteoporosis, arthritis, periodontal disease, and delayed fracture union [3,4,5,6]. These diseases are common worldwide and cause serious social and economic problems. In our previous study, bone invasion was also present in invasive PA [6,7]. However, the exact mechanisms are unclear, and the existing treatments are far from satisfactory. Therefore, it is essential to define new targets and explore the mechanisms underlying osteoclast differentiation and function.

Osteoclasts are bone-specific multinucleate cells derived from the monocyte/macrophage lineage and are activated by various cytokines, including macrophage colony-stimulating factor (M-CSF) and RANKL [1,8]. Under pathological conditions, abnormal osteoclast differentiation and activation often trigger osteolysis. A better understanding of the detailed mechanisms of osteoclast differentiation would greatly aid the development of novel therapeutics targeting pathological osteolysis. During osteoclast maturation, RANKL binds to its receptor RANK, stimulating RANK trimerization and its subsequent association with several tumor necrosis factor receptor-associated factors (TRAFs), of which TRAF6 is one of the most important genes [9]. The ubiquitination of TRAF6, which serves as a bridge between RANK signaling and NF-kB signaling, is essential for proper functioning. CYLD acts as a deubiquitinating enzyme and can prevent RANK-mediated osteoclastogenesis by deubiquitinating TRAF6 [10]. Multiple membrane-initiated signaling pathways are activated, including the MAPK pathway, the NF-кB pathway, and the phosphatidylinositol 3-kinase (PI3K) pathway [11,12,13]. These pathways trigger the expression of downstream transcription factors that promote osteoclastogenesis, including tartrate-resistant acid phosphatase (TRAP), the nuclear factor of activated T cells c1 (NFATc1), activating protein-1 (AP-1), cellular protooncogene Fos (c-Fos), and cathepsin K (CK), thus further driving osteoclast differentiation [14,15].

MicroRNAs (miRNAs) are a group of small noncoding RNAs of approximately [16,17,18,19,20] nucleotides that regulate gene expression at the post-transcriptional level [16]. Members of the miR-19 family (including miR-19a, miR-19b-1, and miR-19b-2), as members of the miR-17/92 cluster, have been identified as important oncogenic miRNAs that have been reported to be upregulated in a variety of tumors [21,22,23]. Furthermore, several studies have shown that miR-19a/b participates in the physiological and pathological processes of bone metabolism. Taipaleenmäki et al. pointed out that in postmenopausal osteoporosis, miR-19a/b can promote osteoclast activation by activating the NF-κB pathway, leading to bone loss [17]. In addition, miR-19b has also been reported to promote bone loss after spinal cord injury by enhancing autophagy in osteoblasts [18]. These pieces of evidence suggest miR-19a/b are involved in osteoclast differentiation and bone resorption, and considering this, we investigated whether miR-19a (part of miR19a/b) participates in the process of bone invasion in PA. Referring to microRNA microarray data, miR-19a is the up-regulated miR-19 family member in invasive PA, which was most researched in our study. Here, we explored whether miR-19a is involved in RANKL-induced osteoclastogenesis. We found that PA-derived miR-19a promotes osteoclastogenesis by inhibiting the expression of the deubiquitinating enzyme CYLD, suggesting that miR-19a regulates bone resorption.

## 2. Materials and Methods

### 2.1. Fluorescence In Situ Hybridization (FISH)

Patient tumor samples were obtained from the Neurosurgery Department of Tongji Hospital, all patients gave informed consent, and patient characteristics are provided in Appendix A. Pituitary tumor tissue was embedded and sectioned by Biossci (Wuhan, China). Slices were processed until antigen repair, then digested with proteinase K (20 μg/mL) for 20–30 min at 37 °C, followed by the addition of pre-hybridization solution, their incubation at 37 °C for 1 h, and the addition of probe-containing hybridization solution; miR-19a bioprobes were designed by Servicebio (Wuhan, China). Then, slices were carried out overnight in a thermostat at 37 °C. Photographs were taken after staining the nuclei with DAPI.

### 2.2. Animal Experiments

Animal experiments were performed according to the experimental protocol approved by the Animal Protection Committee of Tongji Hospital (No. TJH-202206015). Male BALB/c nude mice (Gempharmatech, Nanjing, China) used were housed in an animal facility under SPF conditions. These mice were used to establish a tumor bone invasion model of PA at 6–7 weeks of age; GH3 in PBS (1 × 10^6^ cells/100 µL) were injected into the mice between the muscle layer and the periphery of the skull, above the skull. All animals were killed on day 15 after being injected four times with miR-19a antagomir (5 nmol/100 μL) or vehicle. Tumor and skull specimens were surgically collected and fixed in 4% paraformaldehyde for 48 h and decalcified in 10% EDTA for 15 days. The specimens of tumor and skull were embedded in paraffin, and the 5 μm thick coronal sections were prepared and stained by HE. Based on the results of HE staining, a score for bone erosion was assigned to each bone invasion specimen.

### 2.3. Cell Culture and Reagents

Murine bone marrow-derived macrophages (BMMs) were isolated, as described previously [24]. In brief, bone marrow cells were flushed from the femora and tibiae of 8-week-old C57BL/6 mice and cultured for 2 days with complete α-MEM medium containing 10% (*v*/*v*) fetal bovine serum, 100 U/mL penicillin, 100 µg/mL streptomycin, and 30 ng/mL M-CSF. Cells in the supernatant were collected and cultured in the above medium for 2 days. Adherent cells were considered to be BMMs. GH3 and 293T cells were cultured in completed DMEM supplemented with 10% fetal bovine serum. A miR-19a mimic and a scrambled control were synthesized by Tsingke Biotechnology Company (Beijing, China). Lentivirus expressing CYLD and scrambled control virus were purchased from Vigene Biosciences (Rockville, MD, USA) and transfected by Lipofectamine 3000 transfection reagent (Thermo Fisher Scientific, Waltham, MA, USA). Recombinant mouse M-CSF and RANKL were purchased from R&D Systems (Minneapolis, MN, USA). Primary antibodies included anti-GAPDH, CYLD (ProteinTech Group, Chicago, IL, USA), TRAF6, TRAP, K63-specific polyubiquitin, NFATc1, c-Fos, AP-1, p65, p-p65, NF-кB, p38, p-p38, JNK, p-JNK, ERK, and p-ERK (Cell Signaling Technology, Beverly, MA, USA). 

### 2.4. Osteoclastogenesis Assay

To generate osteoclasts, primary BMM cells were stimulated with 30 ng/mL M-CSF and 50 ng/mL RANKL in medium for 5 days, which was changed every 24 h. For TRAP staining, cells were fixed with 4% (*w*/*v*) paraformaldehyde and stained by the manufacturer (TRAP staining kit, Sigma-Aldrich, St. Louis, MO, USA). TRAP-positive cells with more than three nuclei were identified as osteoclasts.

### 2.5. F-Actin Ring Assay

Osteoclasts are induced from BMM under RANKL stimulation, as described previously [25], fixed with 4% (*v*/*v*) formaldehyde for 10 min, permeabilized with 0.1% (*v*/*v*) Triton X-100 for 10 min, incubated with rhodamine-conjugated phalloidin (Sigma-Aldrich, St. Louis. MO, USA) for 30 min, then with secondary antibodies for 1 h at room temperature, and stained with DAPI to visualize the nuclei. Images were captured using the Nikon Eclipse 80i microscope (Nikon, Tokyo, Japan).

### 2.6. Bone Resorption Assay

BMMs were added to 12-well plates (Corning, New York, NY, USA) at 40,000 cells/well. After 5 days of culture with M-CSF and RANKL, mature osteoclasts were collected and planted in a Corning OsteoAssay-surfaced 96-well plate. The cells were transfected with the miR-19a mimic or vector and cultured for 3 days; the plate was then washed with 10% (*v*/*v*) bleaching solution for 5 min. Pit formation was viewed under a light microscope and quantified.

### 2.7. Quantitative Real-Time PCR (qRT–PCR)

Cultured cells were harvested, and total RNA was extracted using Trizol reagent (Invitrogen, Carlsbad, CA, USA). The RevertAid First Strand cDNA Synthesis Kit (Thermo Scientific, Waltham, MA, USA) was utilized to synthesize first-strand cDNA. qRT-PCR was performed using Fast SYBR Green Master Mix (Thermo Fisher Scientific), carried out on the iCycler real-time PCR instrument (Bio-Rad, Carlsbad, CA, USA). The levels of mRNA expression were normalized to the internal reference gene GAPDH, and the miRNA level was normalized to U6. All primers were synthesized by Tsingke Biotechnology Company (Beijing, China). The primer sequences were (forward; reverse): U6, F 5′-GCTTCGGCAGCACATATACTAAAAT-3′ and R5′-CGCTTCACGAATTTGCGTGTCAT-3′; GAPDH, F 5′-CTCCCACTCTTCCACCTTCG-3′ and R5′-TTGCTGTAGCCGTATTCATT-3′; TRAP, F 5′-GATGCCAGCGACAAGAGGTT-3′ and R 5′-ATACCAGGGGATGTTGCGAA-3′; NFATc1, F 5′-CAACGCCCTGACCACCGATAG-3′ and R 5′-GGGAAGTCAGAAGTGGGTGGA-3′; CK, F 5′-GAAGAAGACTCACCAGAAGCAG-3′ and R 5′-TCCAGGTTATGGGCAGAGATT-3′; CYLD, F 5′-GGATAACCCTATTGGCAACTGG-3′ and R 5′-TTGGAAGTCCCTGGGATGATG-3′; TRAF6 F 5′-AAAGCGAGAGATTCTTTCCCTG-3′ and R 5′-ACTGGGGACAATTCACTAGAGC-3′; miR-19, F 5′-GCAGTGTGCAAATCTATGCAA-3′ and R 5′-GGTCCAGTTTTTTTTTTTTTTTCAGT-3′.

### 2.8. Dual-Luciferase Reporter Gene Assay

The CYLD 3ʹ-UTR was amplified by PCR using the following primers: F 5′-GAGCTCGGCACCCATTGCCGGCA-3′ and R 5′-TCTAGAGGCAATCATTAGCTACA-3′. The purified PCR product was inserted into the pmirGLO dual luciferase plasmid to construct miRNA target expression vector (Promega, Madison, WI, USA). Moreover, 293T cells were planted into 96-well plates and transfected with the pmirGLO-CYLD 3′-UTR and either the miR-19a mimic or the control vector. After 48 h, the supernatants were collected, and the dual luciferase reporter kit (Promega, Madison, WI, USA) was used to detect the luciferase activity. Analysis of the firefly luciferase values was normalized to Renilla luciferase values to acquire the binding activity of microRNA.

### 2.9. Western Blotting (WB) and Co-Immunoprecipitation (Co-IP)

The WB procedure has been described [26]. Proteins were extracted from lysed cultured cells by RIPA buffer containing 1 mM phenylmethanesulfonylfluoride (PMSF). Equal amounts of protein were separated by electrophoresis in 10% (*w*/*v*) SDS–PAGE and transferred to PVDF membranes (Millipore, Billerica, MA, USA). The membranes were blocked with 5% (*w*/*v*) BSA for 1 h at room temperature and then incubated with primary antibodies at 4 °C overnight, followed by washing with TBS containing 0.1% (*v*/*v*) Tween 20. The blots were incubated with HRP-conjugated secondary antibodies (ProteinTech Group, Chicago, IL, USA) at 37 °C for 1 h with gentle shaking, after which ECL reagent (Thermo Fisher Scientific) was added. For co-immunoprecipitation, total proteins from BMM were pre-cleared by incubating with the control IgG and protein A/G beads for 1 h to remove proteins that were nonspecifically bound to the immunoprecipitation components. The antibodies of anti-TRAF6 and protein A/G beads were incubated with the pre-cleared cell lysates overnight. The immunoprecipitated protein was washed 3 times with lysis buffer and then subjected to WB after boiling. For each WB result, at least three replications were quantitated and analyzed with ImageJ (version 1.53T).

### 2.10. Electrophoretic Mobility Shift Assay (EMSA)

After being transfected with miR-19a or vector for 24 h, BMMs were stimulated with or without RANKL (75 ng/mL) for 30 min. Nuclear proteins were extracted using a nuclear/cytoplasmic protein extraction kit (Beyotime Institute of Biotechnology, Jiangsu, China). The nuclear extracts were incubated with NF-кB-specific probe or AP1-specific probe (Beyotime Institute of Biotechnology) at room temperature for 30 min. Then, the proteins were separated on a 6% (*w*/*v*) polyacrylamide gel, transferred to the Amersham Hybond-N+ membrane (GE Healthcare, Little Chalfont, UK), and cross-linked under UV. The membrane was then blocked and incubated with a streptavidin–horseradish peroxidase conjugate. ECL signals were detected using the ChemiDoc XRS+ System (Bio-Rad). 

### 2.11. Statistical Analysis

Experiments in the research were performed repetitively at least three times. The unpaired Student’s *t*-test was used for statistical comparisons. One-way analysis of variance (ANOVA) was used to determine significant differences in multiple comparisons. A *p*-value < 0.05 was considered statistically significant. Analyses and statistical graphing were performed using GraphPad PRISM v7.0 software or R4.2.1.

## 3. Results

### 3.1. miR-19a Promotes PA Bone Invasion

In the available transcriptome sequencing data of PA, the miR-19 expression was significantly upregulated in the invasive PA group (Figure 1A). By FISH staining, the miR-19a expression was higher in PA tissues with bone invasion compared to non-invasive PA (Figure 1B). In a nude mouse model of calvaria xenograft, PA tumors overexpressing miR-19a showed a more pronounced bone invasive effect, but there was no significant difference in tumor size between the groups (Figure 1C,D). However, the degree of bone invasion was significantly attenuated after the intratumoral injection of miR-19a antagomir (Figure 1D), suggesting that miR-19a not only affects tumor cells but may function on other cells, thereby promoting bone invasion.

### 3.2. miR-19a Accelerates Osteoclast Differentiation and Function

To explore any role of miR-19a in osteoclastogenesis, miR-19a expression in RANKL-induced BMMs was determined by qPCR. During RANKL-induced osteoclast differentiation, miR-19a expression gradually increased over time (Figure 2A). We next explored whether miR-19a affects osteoclast differentiation. BMMs were transfected with miR-19a or the control followed by M-CSF and RANKL to stimulate differentiation into osteoclasts. TRAP staining (Figure 2B) showed that osteoclast numbers increased significantly 5 days after stimulation in the presence of miR-19a overexpression.

An F-actin ring assay and pit formation assay were applied to investigate the effect of miR-19a on osteoclast resorption. After transfection with an miR-19a mimic or the control for 2 days, osteoclast development was then induced (in a 12-well plate) through incubation with M-CSF and RANKL for 3 days. The formation of F-actin was assessed, and we observed a significant increase after treatment with miR-19a (Figure 2C). Additionally, miR-19a significantly enhanced pit formation (Figure 2D) and upregulated osteoclast resorption. To further prove the effects of miR-19a on osteoclastogenesis, we measured the expression levels of the osteoclast-related transcription factors NFATc1, c-Fos, TRAP, and cathepsin K after miR-19a transfection. qPCR results revealed that all genes were significantly upregulated by miR-19a (Figure 2E), which is consistent with Western Blot analysis (Figure 2F), thereby promoting RANKL-induced osteoclast differentiation (Figure 2G). 

### 3.3. miR-19a Inhibits CYLD Expression and, Thus, Promotes TRAF6 Ubiquitination 

To clarify how miR-19a affects osteoclast formation, The target genes of miR-19a were predicted by the TargetScan database (http://www.targetscan.org (accessed on 15 December 2022)). The NF-кB pathway is one of the major pathways involved in osteoclastogenesis, and in the overall set of NF-кB pathway genes (GSEA standard name: PID_NFKAPPAB_CONNICAL_PATHWAY) that bind to miR-19, CYLD is the only target gene (Figure 3A). The 3′-UTR of CYLD was cloned into a dual-luciferase reporter vector and co-transfected with the miR-19a mimic or the control into 293T cells. A dual luciferase assay indicated that miR-19-3p bound to the conserved 3′-UTR sites of CYLD mRNA and inhibited CYLD promoter activity (Figure 3D,E). qPCR and WB showed that CYLD mRNA and protein levels (Figure 3B,C), respectively, were significantly decreased by miR-19. The activity of TRAF6 to induce the NF-κB pathway is dependent on the K63 ubiquitination of TRAF6. Considerable research suggests that CYLD mainly inhibits the ubiquitination of TRAF6 [10,27,28]. The co-immunoprecipitation results indicated the binding of CYLD to TRAF6 (Figure 3F,G). To sum up, miR-19a upregulated the ubiquitination of TRAF6 K63 by inhibiting CYLD.

### 3.4. CYLD Is Required for miRNA-Mediated Regulation of Osteoclast Formation

We wanted to investigate whether the overexpression of CYLD could reverse the effect of osteoclast differentiation mediated by miR19. We infected BMMs with lentivirus overexpressing CYLD (or empty vector) after transfection with miR-19a (Figure 4A). The CYLD overexpression in BMMs blocked miR-19-mediated osteoclastogenesis by TRAP staining (Figure 4B).

### 3.5. miR-19a Promotes RANKL-Induced NF-кB and p-ERK Expression

The RANKL binding to RANK in osteoclast precursor cells activates the downstream NF-кB and MAPK pathways [1]. An electrophoretic mobility shift assay revealed that miR-19a dramatically increased NF-кB and p-ERK DNA-binding activity (Figure 5A). Notably, NF-ATc1 and AP-1 induction were promoted by miR-19. MAPK and NF-кB activation were enhanced principally by the upregulation of ERK and p65 phosphorylation after the miR-19a overexpression (Figure 5B,C).

## 4. Discussion

Osteoclasts are the major cells involved in bone resorption and remodeling. Bone destruction induced by overactivated osteoclasts causes many diseases. The pathogenetic mechanism of osteoclast formation is not fully understood. Our previous study found that bone erosion is also present in invasive PA [6], but the function that osteoclasts play in PA is unclear. 

Studies have shown that miRNAs, especially the miR-17/92 cluster, can influence the process of bone remodeling or tumor bone metastasis by regulating gene expression. For example, in lung cancer, miR-17 promotes osteoclast differentiation by targeting PTEN and mediates tumor bone metastasis [22]. miR-20a inhibits hypoxia-induced osteoclast differentiation by regulating the expression of autophagy-related proteins [29], and miR-18a-3p promotes osteoporosis and fracture by inhibiting the glutamate AMPA receptor subunit 1 gene [30]. 

miR-19, as a subfamily of the miR-17/92 cluster, plays essential roles in many diseases, potentiating NF-кB activity in inflammation and promoting the malignancy of osteosarcoma as well as regulating bone metabolism, which have been reported in several studies. Studies have demonstrated that miR-19a/b mediates osteoporosis in the postmenopausal period or after spinal cord injury by promoting osteoclast differentiation [17,18]; in addition, some have shown that miR-19a/b promotes osteoblast activation as well [31,32,33]. Tumor metastasis and invasion into bone tissue are dependent on the remodeling of the bone structure, and the involvement of miR-19a/b in this process has been reported in prostate cancer and breast cancer [23,33]. We found that the PA-derived miR-19a promoted bone invasion in PA. The miR-19a level gradually increased during RANK activation, and miR-19a significantly promoted osteoclastogenesis. NFATc1, c-Fos, AP-1, and TRAP are hallmarks of osteoclast differentiation, and their abnormal expression may contribute to serious bone diseases. We found that the mRNA and protein expression levels of NFATc1, c-Fos, and TRAP were upregulated by miR-19, indicating that miR-19a enhances RANKL-induced osteoclastic differentiation in vitro.

CYLD is usually considered to be a tumor suppressor gene that inhibits TRAF6 ubiquitination by binding to P62, thus modulating the entire RANKL/RANK/TRAF6 pathway [10]. CYLD regulates the activation of ERK, IKK, and NF-кB by affecting the ubiquitination of TRAF2 and TRAF6 [34,35]. The important role of CYLD in osteoclastogenesis has been well-established [10,36]. CYLD negatively regulates osteoclast RANK signaling by inhibiting TRAF6 ubiquitination. We found that miR-19a downregulated the expression of CYLD in BMM and increased TRAF6 ubiquitination, as reported in previous studies [10,37]. CYLD overexpression prevented the effects of miR-19. Thus, miR-19a promoted osteoclast differentiation by suppressing CYLD expression and increasing TRAF6 ubiquitination. When RANKL binds to its membrane receptor RANK, the cytoplasmic domain of the receptor recruits TRAF6 to activate downstream NF-кB and MAPK signaling [38], which are considered major pathways in osteoclastogenesis. 

Still, there are several limitations in this study. First, we have only investigated the miR-19a in the process of PA-mediated bone invasion, while other candidates can also be participators. Additionally, the sample size of the public miRNA-seq dataset is very limited. Therefore, a large cohort is needed to comprehensively present miRNA profiles in bone invasive PA. In addition, the sample size of our collected clinical specimens is also relatively limited, which requires a larger cohort to systematically validate our findings from a clinical perspective.

In conclusion, we demonstrated that PA enhances the process of bone invasion by upregulating miR-19a function on osteoclasts. It remains to be further verified whether other members of the miR-19 family have similar roles, although we speculate that they may have similar functions based on similar mature sequences. Our study was directed only in pituitary adenomas, and it is not clear whether its conclusions can be extended to other tumors or bone metabolic diseases.

## 5. Conclusions

In summary, we found that miR-19a promoted osteoclast differentiation and regulated bone resorption while mediating bone invasion in PA. Surprisingly, we observed that CYLD is a target of miR-19. miR-19a increased the K63 ubiquitination of TRAF6 via CYLD. Further results indicated that miR-19a promotes osteoclastogenesis via the activation of the downstream NF-кB and MAPK pathways. Finally, we propose a mechanistic model in which miR-19a promotes osteoclastogenesis by inhibiting CYLD expression and enhancing the activation of the NF-кB and MAPK pathways. Given our previous studies related to bone invasion as well as osteoporosis in PA patients, our study also provides guidance on the genesis of osteoclast-related disorders caused by PA patients in clinical settings.

## Figures and Tables

**Figure 1 cancers-16-00302-f001:**
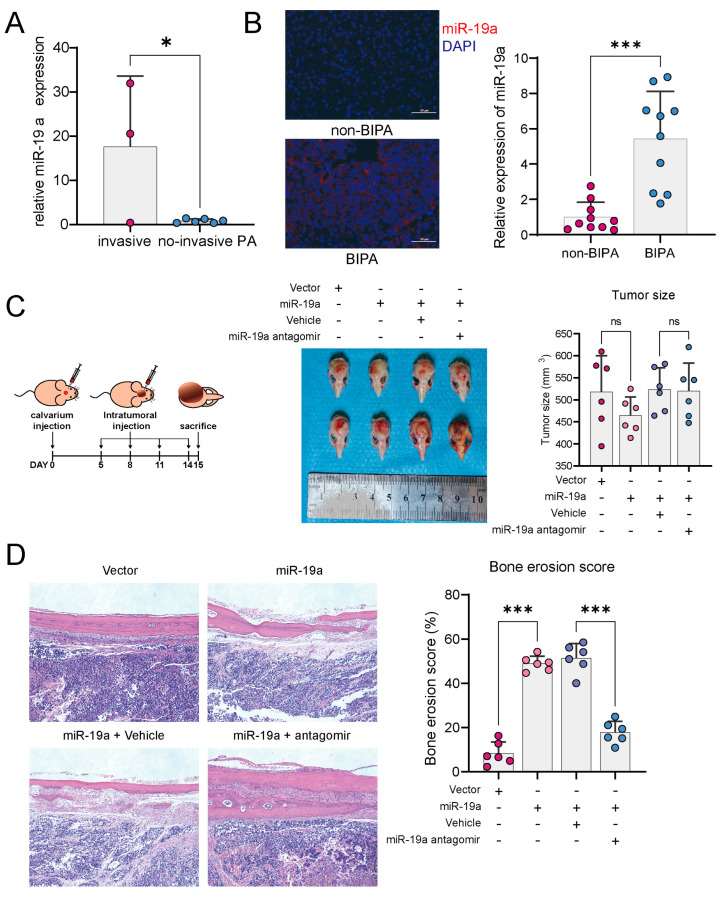
miR−19a promotes bone invasion in PA. (**A**) miR-19a expression levels in transcriptome sequencing data from GEO database (GSE46294) between invasive and non-invasive PAs. (* *p* < 0.05). (**B**) FISH of miR-19a in patients between bone-invasive PA(BIPA) and non-BIPAPAs. (blue, DAPI; red, miR-19a) Quantification of relative miR-19a expression, right (*n* = 10 mice per group; *** *p* < 0.001) (BIPA, bone invasive pituitary adenoma). (**C**) Schematic diagram of calvaria xenograft PAs in nude mice, left. Photographs of GH3 calvaria xenograft tumor specimens, middle. Quantification of tumor size, right (*n* = 6 mice per group; ns *p* ≥ 0.05). (**D**) HE staining exhibited bone destruction of GH3 tumors. Quantification of bone erosion score, right (*n* = 6 mice per group; *** *p* < 0.001).

**Figure 2 cancers-16-00302-f002:**
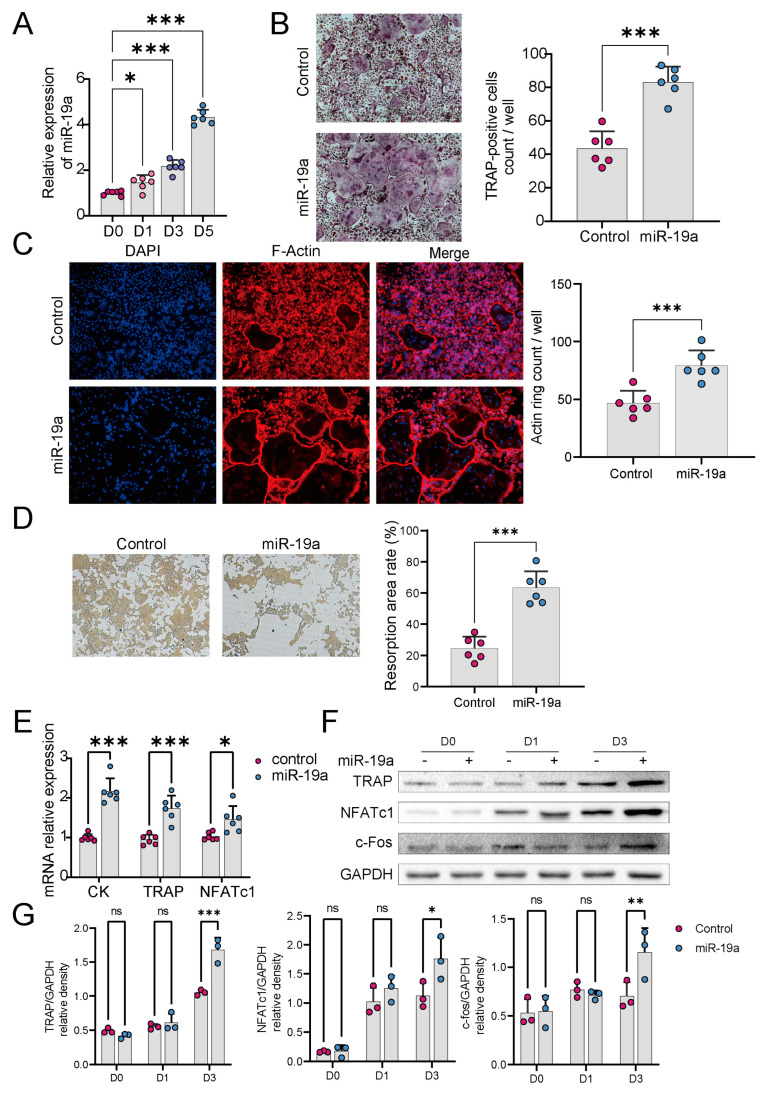
miR−19a promotes osteoclast differentiation and function. (**A**) PCR of miR-19a in BMMs, which were stimulated with 50 ng/mL RANKL for 1, 3, and 5 days (*n* = 6 independent experiments; *** *p* < 0.001; * *p* < 0.05). (**B**) TRAP staining of BMMs transfected with miR-19a or scramble control (×200). Quantification, right (*n* = 6 independent experiments; *** *p* < 0.001). (**C**) F-actin staining assay of BMMs transfected with miR-19a or scramble control (×200). Quantification, right (*n* = 6 independent experiments). (**D**) pit formation assay of BMMs transfected with miR-19a or scramble control (×200). Quantification, right (*n* = 6 independent experiments; *** *p* < 0.001). (**E**) qPCR of osteoclast-related marker genes (CK, TRAP, NFATc1) in BMMs transfected with miR-19a or scramble control (*n* = 6 independent experiments; *** *p* < 0.001; * *p* < 0.05). (**F**,**G**) WB of osteoclast-related marker genes (CK, TRAP, NFATc1) in BMMs transfected with miR-19a or scramble control at 0, 1, and 3 days (**F**). Quantification of WB (**G**). (*n* = 3 independent experiments; *** *p* < 0.001; ** *p* < 0.01; * *p* < 0.05; ns *p* ≥ 0.05). The original images of the Western Blotting figures can be found in Appendix A.

**Figure 3 cancers-16-00302-f003:**
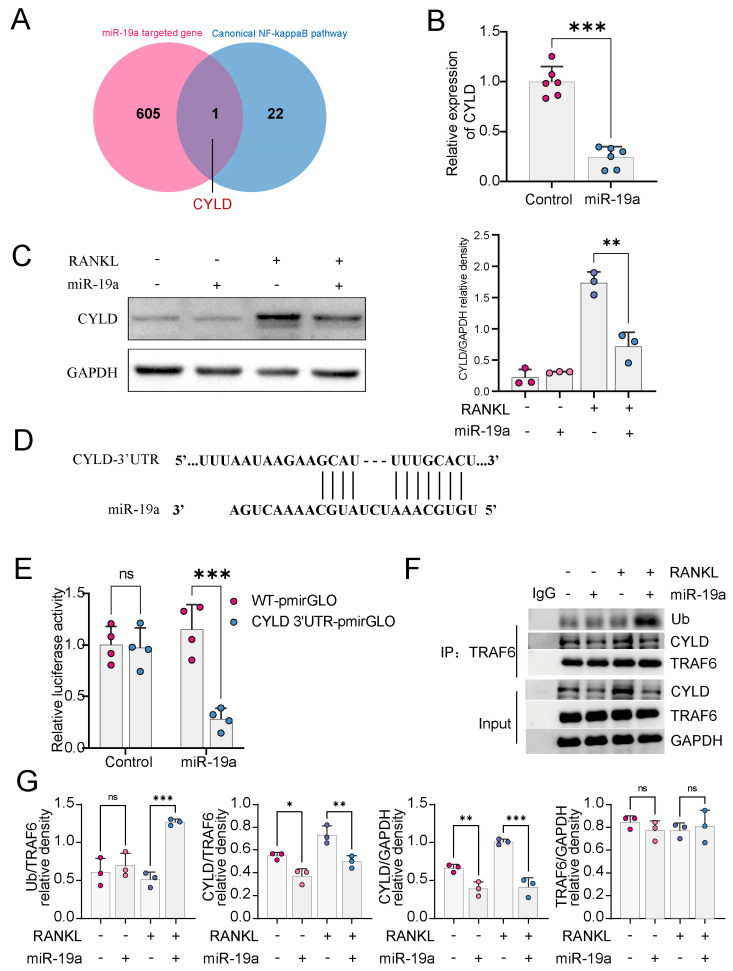
miR−19a inhibits CYLD expression and, thus, promotes TRAF6 ubiquitination. (**A**) Venn diagram demonstrating the intersection of miR-19a binding mRNA and NF-κB pathway-related genes. (**B**) qPCR of CYLD in BMMs transfected with miR-19a or scramble control (*n* = 6 independent experiments; *** *p* < 0.001). (**C**) WB of CYLD in BMMs transfected with miR-19a or scramble control. Quantification, right (*n* = 6 independent experiments; ** *p* < 0.01). (**D**) The complementary sequences of miR-19-3p at the 3′UTR of CYLD mRNA were identified in TargetScan. (**E**) Dual luciferase assay for 293T to detect the binding ability of miR-19a to 3′UTR of CYLD mRNA (*n* = 4 independent experiments; *** *p* < 0.001; ns *p* ≥ 0.05). (**F**,**G**) Co-IP assay to obtain TRAF6 binding protein for WB detection of ubiquitin and CYLD. (**F**) Quantification of WB. (**G**) (*n* = 3 independent experiments; *** *p* < 0.001; ** *p* < 0.01; * *p* < 0.05; ns *p* ≥ 0.05). The original images of the Western Blotting figures can be found at Appendix A.

**Figure 4 cancers-16-00302-f004:**
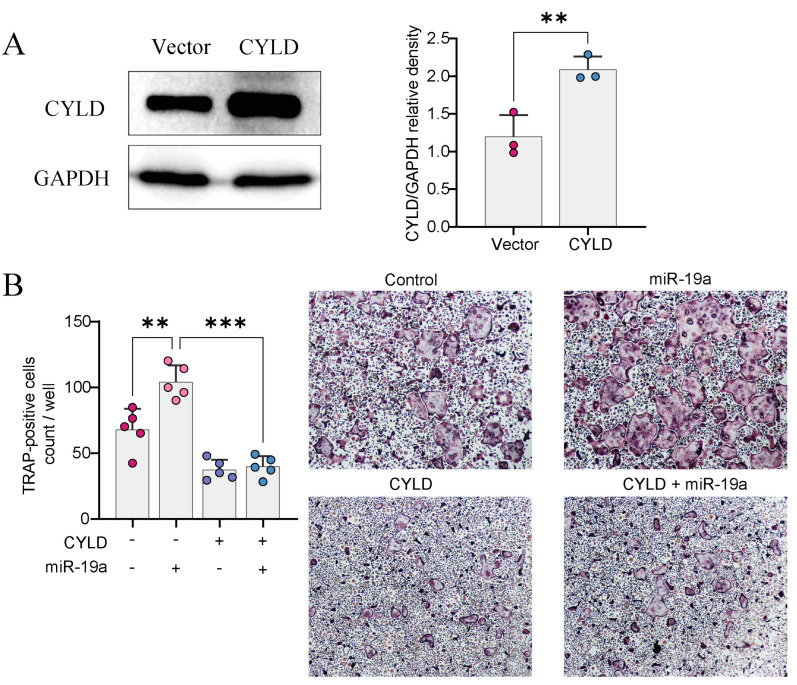
Overexpressing CYLD reverses the effect of miR-19a on osteoclastogenesis. (**A**) WB detection of CYLD overexpression efficiency in BMMs. Quantification, right (*n* = 3 independent experiments; ** *p* < 0.01). (**B**) TRAP staining of BMMs transfected with miR-19a or CYLD overexpression, right (×200). Quantification, left (*n* = 5 independent experiments; *** *p* < 0.001; ** *p* < 0.01). The original images of the Western Blotting figures can be found in Appendix A.

**Figure 5 cancers-16-00302-f005:**
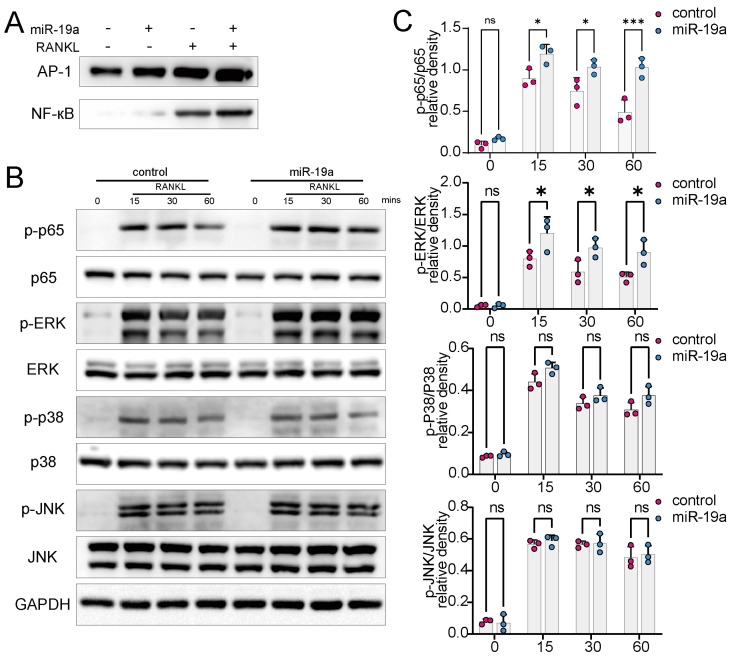
NF-кB and MAPK/ERK signaling are activated after transfection of miR-19. (**A**) EMSA assay to detect DNA-binding activity of AP1 and NF-κB after transfection with miR-19a mimic or RANKL administration. (**B**,**C**) WB assays to detect NF-кB and MAPK/ERK signaling after transfection with miR-19a mimic or RANKL administration (B). Quantification of WB (**C**) (*n* = 3 independent experiments; *** *p* < 0.001; * *p* < 0.05; ns *p* ≥ 0.05). The original images of the Western Blotting figures can be found in Appendix A.

## Data Availability

The datasets in this study are available from the corresponding author upon reasonable request.

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
