# Peer review of "The miR-19a/Cylindromatosis Axis Regulates Pituitary Adenoma Bone Invasion by Promoting Osteoclast Differentiation"

_cancers, 2024, doi:10.3390/cancers16020302_

Round 1

Reviewer 1 Report

Comments and Suggestions for Authors

In this study, the author examined that the effects of transfection of miR-19 on RANKL-induced osteoclastogenesis in bone marrow-derived macrophages. These are based on the findings that miR-19 level was higher in patients of pituitary adenoma with bone invasion than without bone invasion. The transfection of miR-19 induced osteoclastogenesis in the presence of RANKL via suppressing of CYLD expression. NF-кB and MAPK that are associated RANKL activation were also enhanced by miR-19. Therefore, the authors concluded miR-19 induced osteoclast differentiation, and these effects of miRNA were involved in bone invasion in PA.    

1) If the results shown in Figures 1A and B were obtained from samples of PT patients, it is necessary to indicate the background information of patients (ages, gender, etc.). Moreover, it is also required to describe obtaining informed consent from patients and ethical approval in the research institution.

2) Figure 2A and Figure 2C indicted that RANKL induced not only miRNA-19 expression but also CYLD expression which is a the target of miR-19. Besides, in lines 359-364, the author indicated that CYLD negatively regulates osteoclastogenesis. How should we understand this conflict in RANKL-induced osteoclastogenesis?

Author Response

In this study, the author examined that the effects of transfection of miR-19 on RANKL-induced osteoclastogenesis in bone marrow-derived macrophages. These are based on the findings that miR-19 level was higher in patients of pituitary adenoma with bone invasion than without bone invasion. The transfection of miR-19 induced osteoclastogenesis in the presence of RANKL via suppressing of CYLD expression. NF-кB and MAPK that are associated RANKL activation were also enhanced by miR-19. Therefore, the authors concluded miR-19 induced osteoclast differentiation, and these effects of miRNA were involved in bone invasion in PA.    

1) If the results shown in Figures 1A and B were obtained from samples of PT patients, it is necessary to indicate the background information of patients (ages, gender, etc.). Moreover, it is also required to describe obtaining informed consent from patients and ethical approval in the research institution.

Response: We will provide the patient's informed consent form, ethical approval and basic patient information (see Supplementary Table 1) as an attachment.

2) Figure 2A and Figure 2C indicted that RANKL induced not only miRNA-19 expression but also CYLD expression which is a the target of miR-19. Besides, in lines 359-364, the author indicated that CYLD negatively regulates osteoclastogenesis. How should we understand this conflict in RANKL-induced osteoclastogenesis?

Response: RANKL is a classical cytokine that promoting osteoclast differentiation. Current studies suggest that RANKL promotes osteoclastogenesis mainly by activating the NF-κB pathway in osteoclasts. In addition, it has been widely recognized that the activation of the NF-κB pathway is accompanied by the up-regulation of the expression of CYLD, which is a “negative” feedback regulator of NF-κB. According to the miRNA-mRNA sponging theory, our study shows that miR-19 indeed can target and “suppress”, instead of elevating, CYLD expression, which attenuate CYLD-mediated negative feedback regulation for NF-κB pathway, and thus promote RANKL-induced osteoclastogenesis.

Reviewer 2 Report

Comments and Suggestions for Authors

The study provides evidence of miR-19-CYLD axis involvement in osteoclast differentiation and bone degradation with relevance in the context of bone invasion by pituitary adenomas. While the study provides important results for this field of oncology, I have major concerns that the authors need to address before considering this study for publication.

Major point:

1)      Throughout the whole study to which miR-19 are you referring? miR-19a-3p or miR-19b-5p? Please replace everywhere it appears miR-19 to which one exactly are you referring.

2)      The introduction and discussion sections contain redundant information. The role of miR-17/92 family is written in introduction as well as in discussions. I would focus more on just the roles of mIR-19 in bone metabolism and bone metastasis in other diseases (in the Discussion section). There are several studies on this subject.

3)      There is a major focus on the miRNAs, but please add more information, from literature, in introduction or discussion what does CYLD do in physiological and pathological context.

4)      The Discussion sections needs to be written very differently, please also include study limitations and further perspective in the discussion section

Minor issues:

Page 1, line 26-28, abstract: “miRNAs have been reported to participate in the pathophysiological processes of many 26 diseases, including osteoclast differentiation. miR-19 (as a key member of the miR-17-92 cluster) 27 activates the nuclear factor-кB (NF-кB) pathway and promotes inflammation.” – shorten this phrase or delete it.

Line 92- miR19a or miR-19b? “Remarkably, miR-19 has”

Line 95-97:move it to discussion section. The formulation “What’s more” needs to be changed, it is too informal.

Lines 97-99- I do not see the relevance of talking about what does miR-19 do in rheumatoid arthritis since there is plenty of information about its role in osteoclast/osteoblast differentiation or bone metastasis.

Line 107- change “tumor tissue is embedded” to “tumor tissue was embedded”. Check the tense of all verbs for consistency

Lines 225-226 delete this part, it is redundant to repeat the reasons for testing miR-19. This should be included in the last part of introduction

Lines 226-227 – results present in Figure 1A. The sample size is too small to include this data and draw a reliable conclusion. From what I can see there were only 3 samples for invasive PA, please include the limitations of these results, improve them or delete this part

Lines 238-239 include also in Figure 1B legend that DAPI is on blue and miR-19 (a or b?) on red

Lines 239-240- There are several problems with 1C: the images of tumors are hard to see and which one belongs to which group is hard to discern, the graph: miR-19 contains miR-19 mimic and antagomir. how do you explain the fact that miR-19 group seems to have a smaller tumor size. Please provide the statistical tests results from the comparison between miR-19 and vehicle or miR-19 antagomir.

Line 240-241: in the figure legend only explain what it is seen in the picture.

Line 251-253: The transfection took place before or after M-CSF and RANKL incubation?

Line 254: were treated with mimic for 3 days everyday or were treated and then evaluated after 3 days?

Line 262- Figure2 legend, there is too much information given  about the methods. Just explain what is in the picture.

Line 271, (D): this is information related to methods and not to what it is seen in the picture. example: Box plots and pictures of the pit assay comparison between control and miR-19 mimic treated cells. what statistical test was applied, what does *** mean?

Lines 289-291: “TRAF6 expression by inducing ubiquitination of TRAF6 K63.–  rephrase this sentence

Figure 3A: the whole list is not needed in Figure 3A,( the list of genes with CYLD highlighted) just mention which is the common gene

Line 302: correct “showedthat”

Line 306- Please explain better. while in Figure 3F there is an increase in Ub I did not observe a difference of TRAF6 level. Can you provide the densitometry analysis and statistical test that proves an increase in TRAF6 level after miR-19 mimic treatment?

Line 338- “Many studies have shown that miRNAs are important regulators of gene 338 expression” – move it to a new paragraph and also include that miRNAs directly target gene expression and thus regulate bone formation and breakdown.

Lines 339-349- this is information similar to that of the introduction. There are too many examples here and it is irrelevant to this subject. shorten this part and focus on miR17/92 cluster and its role in bone formation.

Line 352: “but any effect of miR-19 on bone metabolism was unclear.”- this is not true. There are some studies that focus on the role of mIr-19 in bone formation (https://www.nature.com/articles/s12276-021-00631-w  , https://pubmed.ncbi.nlm.nih.gov/36193848/ , https://onlinelibrary.wiley.com/doi/full/10.1111/jcmm.16159 , https://bmcmedgenet.biomedcentral.com/articles/10.1186/s12881-020-0948-y , https://www.spandidos-publications.com/10.3892/or.2017.6096) – please cite these

Line 368 “The NF-кB and MAPK signaling pathways play major roles in RANKL-induced 368 osteoclastogenesis. “ – move this to the end of the phrase

Line 372: “topositively” correct

Line 373: correct “ourstudy”

Line 375: delete Fig.6 , there is no need to refer to figures here

Lines 377: “, but the effect of miR-19 on osteoblast 377 differentiation remains unknown and requires further investigation.” – there are studies provided some information regarding the roles of mIR-19 in osteoblasts differentiation , please refer to them : https://asbmr.onlinelibrary.wiley.com/doi/full/10.1002/jbm4.10745 , https://pubmed.ncbi.nlm.nih.gov/30136155/ , https://www.ncbi.nlm.nih.gov/pmc/articles/PMC6953218/ , https://journals.plos.org/plosone/article?id=10.1371/journal.pone.0043800 )

Line 383- delete bona fide

Line 384- I did not understand the part about “reduced TRAF6 expression” , because in the Western blot images from Figure 3F I did not observe any difference in the band wide and intensity for TRAF6. – please clarify

Comments on the Quality of English Language

Minor editing is needed. I have found some words that were not separated and  two instances of a too informal language. Overall, the use of English is very good. 

Author Response

The study provides evidence of miR-19-CYLD axis involvement in osteoclast differentiation and bone degradation with relevance in the context of bone invasion by pituitary adenomas. While the study provides important results for this field of oncology, I have major concerns that the authors need to address before considering this study for publication.

Major point:

1)      Throughout the whole study to which miR-19 are you referring? miR-19a-3p or miR-19b-5p? Please replace everywhere it appears miR-19 to which one exactly are you referring.

Response: We studied miR-19a rather than miR-19b-1 and miR-19b-2, including the FISH probes, PCR primers, and microRNA mimic and antagomir we used in this research. The mature sequence of functioning is miR-19a-3p. We have now revised the terms in the text according to the reviewer’s suggestion.

2)      The introduction and discussion sections contain redundant information. The role of miR-17/92 family is written in introduction as well as in discussions. I would focus more on just the roles of mIR-19 in bone metabolism and bone metastasis in other diseases (in the Discussion section). There are several studies on this subject.

Response: According to the reviewer’s suggestion, we have now revised and added the sentences regarding roles of mIR-19 in bone metabolism and bone metastasis in the INTRODUCTION and DISCUSSION. We thank for the suggestion.

3)      There is a major focus on the miRNAs, but please add more information, from literature, in introduction or discussion what does CYLD do in physiological and pathological context.

Response: We have now revised the INTRODUCTION and added related information about CYLD as suggested by the reviewer.

4)      The Discussion sections needs to be written very differently, please also include study limitations and further perspective in the discussion section

Response: We have now improved the discussion and added study limitations and further perspective in the Discussion section.

Minor issues:

Page 1, line 26-28, abstract: “miRNAs have been reported to participate in the pathophysiological processes of many 26 diseases, including osteoclast differentiation. miR-19 (as a key member of the miR-17-92 cluster) 27 activates the nuclear factor-кB (NF-кB) pathway and promotes inflammation.” – shorten this phrase or delete it.

Response: We have now rephrased the sentences.

Line 92- miR19a or miR-19b? “Remarkably, miR-19 has”

Response: Both miR19a and miR-19b-1 are the members of miR-17-92 cluster.

Line 95-97: move it to discussion section. The formulation “What’s more” needs to be changed, it is too informal.

Response: We have now revised this according to the comment.

Lines 97-99- I do not see the relevance of talking about what does miR-19 do in rheumatoid arthritis since there is plenty of information about its role in osteoclast/osteoblast differentiation or bone metastasis.

Response: We have now revised this according to the comment.

Line 107- change “tumor tissue is embedded” to “tumor tissue was embedded”. Check the tense of all verbs for consistency

 Response: We have now revised this according to the comment.

Lines 225-226 delete this part, it is redundant to repeat the reasons for testing miR-19. This should be included in the last part of introduction

 Response: We have now deleted this according to the comment.

Lines 226-227 – results present in Figure 1A. The sample size is too small to include this data and draw a reliable conclusion. From what I can see there were only 3 samples for invasive PA, please include the limitations of these results, improve them or delete this part

Response: This data is cited from the GEO database (GSE46294) which contains only several samples of pituitary tumors. Indeed, the sample size is limited, therefore we examined additional samples of bone invasive pituitary tumors collected from our center for FISH staining to detect miR-19a levels to verify the findings from RNA-seq datasets. We have now added the limitation of these data in the discussion as suggested by the reviewer.

Lines 238-239 include also in Figure 1B legend that DAPI is on blue and miR-19 (a or b?) on red

Response: We have rewritten the legend.

Lines 239-240- There are several problems with 1C: the images of tumors are hard to see and which one belongs to which group is hard to discern, the graph: miR-19 contains miR-19 mimic and antagomir. how do you explain the fact that miR-19 group seems to have a smaller tumor size. Please provide the statistical tests results from the comparison between miR-19 and vehicle or miR-19 antagomir.

Response: We have now improved our figures. We have enlarged the picture for better illustration. Although the miR-19 group seemed to have a smaller tumor volume, it was not statistically significant after statistical analysis.

The following are the results of the statistical analyses:

Vector VS miR-19a mimic:

P value

0.1917

P value summary

ns

Significantly different (P < 0.05)?

No

One- or two-tailed P value?

Two-tailed

t, df

t=1.400, df=10

Mean of ‘Vector’

517.8

Mean of ‘miR-19 mimic’

465.0

miR-19a mimic + Vehicle VS miR-19a mimic + miR-19a antagomir:

P value

0.9105

P value summary

ns

Significantly different (P < 0.05)?

No

One- or two-tailed P value?

Two-tailed

t, df

t=0.1153, df=10

Mean of ‘miR-19 mimic + Vehicle’

523.6

Mean of ‘miR-19 mimic + miR-19 antagomir’

519.8

Line 240-241: in the figure legend only explain what it is seen in the picture.

Response: We have rewritten the figure legend.

Line 251-253: The transfection took place before or after M-CSF and RANKL incubation?

Response: Mouse bone marrow-derived macrophages were collected from C57BL/6 bone marrow and transfected with miR-19 two days after M-CSF administration. And then administration of M-CSF and RANKL for induction osteoclast differentiation.

Line 254: were treated with mimic for 3 days everyday or were treated and then evaluated after 3 days?

Response: We have rewritten the relevant sentence. BMMs were first transfected with miR-19 mimic for 2 days and subsequently treated with M-CSF/RANKL for 3 days and then evaluated for F-actin.

Line 262- Figure2 legend, there is too much information given  about the methods. Just explain what is in the picture.

Response: We have rewritten the figure legend.

Line 271, (D): this is information related to methods and not to what it is seen in the picture. example: Box plots and pictures of the pit assay comparison between control and miR-19 mimic treated cells. what statistical test was applied, what does *** mean?

Response: We have rewritten the figure legend.

Lines 289-291: “TRAF6 expression by inducing ubiquitination of TRAF6 K63.–  rephrase this sentence

Response: We have now rephrased the sentence.

Figure 3A: the whole list is not needed in Figure 3A,( the list of genes with CYLD highlighted) just mention which is the common gene

Response: We have now re-corrected the picture.

Line 302: correct “showedthat”

 Response: We have now revised this according to the comment.

Line 306- Please explain better. while in Figure 3F there is an increase in Ub I did not observe a difference of TRAF6 level. Can you provide the densitometry analysis and statistical test that proves an increase in TRAF6 level after miR-19 mimic treatment?

Response: It is true that miR19 does not have an effect on the expression level of TRAF6 based on our experiment, but it does have an effect on the level of K63 ubiquitination of TRAF6 (but not ubiquitination mediated degradation, which is widely recognized by other researchers [1,2]), and we have rewritten that statement. We have now added the densitometry analysis and statistical test in the revised figure as suggested by the reviewer.

The following are the results of the WB in Figure 3F statistical analyses:

Vector + RANKL VS miR-19a + RANKL:

P value

0.722

P value summary

ns

Significantly different (P < 0.05)?

No

One- or two-tailed P value?

Two-tailed

t, df

t=0.3825, df=4

Mean of ‘miR-19 mimic + Vehicle’

0.7748

Mean of ‘miR-19 mimic + miR-19 antagomir’

0.8094

Line 338- “Many studies have shown that miRNAs are important regulators of gene 338 expression” – move it to a new paragraph and also include that miRNAs directly target gene expression and thus regulate bone formation and breakdown.

 Response: We have now revised this according to the comment.

Lines 339-349- this is information similar to that of the introduction. There are too many examples here and it is irrelevant to this subject. shorten this part and focus on miR17/92 cluster and its role in bone formation.

 Response: We have shortened the paragraph and mainly focused on miR-17/92.

Line 352: “but any effect of miR-19 on bone metabolism was unclear.”- this is not true. There are some studies that focus on the role of mIr-19 in bone formation (https://www.nature.com/articles/s12276-021-00631-w  , https://pubmed.ncbi.nlm.nih.gov/36193848/ , https://onlinelibrary.wiley.com/doi/full/10.1111/jcmm.16159 , https://bmcmedgenet.biomedcentral.com/articles/10.1186/s12881-020-0948-y , https://www.spandidos-publications.com/10.3892/or.2017.6096) – please cite these

 Response: We have rewritten the relevant section and added these references

Line 368 “The NF-кB and MAPK signaling pathways play major roles in RANKL-induced 368 osteoclastogenesis. “ – move this to the end of the phrase

 Response: We have now revised this according to the comment.

Line 372: “topositively” correct

 Response: We have now revised this according to the comment.

Line 373: correct “ourstudy”

 Response: We have now revised this according to the comment.

Line 375: delete Fig.6 , there is no need to refer to figures here

 Response: We have now deleted this according to the comment.

Lines 377: “, but the effect of miR-19 on osteoblast 377 differentiation remains unknown and requires further investigation.” – there are studies provided some information regarding the roles of mIR-19 in osteoblasts differentiation , please refer to them : https://asbmr.onlinelibrary.wiley.com/doi/full/10.1002/jbm4.10745 , https://pubmed.ncbi.nlm.nih.gov/30136155/ , https://www.ncbi.nlm.nih.gov/pmc/articles/PMC6953218/ , https://journals.plos.org/plosone/article?id=10.1371/journal.pone.0043800 )

Response: We have revised the relevant section.

Line 383- delete bona fide

 Response: We have now deleted this according to the comment.

Line 384- I did not understand the part about “reduced TRAF6 expression” , because in the Western blot images from Figure 3F I did not observe any difference in the band wide and intensity for TRAF6. – please clarify

Response: We have now revised this according to the comment. Indeed there is no significant difference in TRAF6 expression, but there is a change in ubiquitination level of TRAF6. It is true that miR19 does not have an effect on the expression level of TRAF6 based on our experiment, but it does have an effect on the level of K63 ubiquitination of TRAF6 (but not ubiquitination mediated degradation, which is widely recognized by other researchers [1,2]), and we have rewritten that statement.

Comments on the Quality of English Language

Minor editing is needed. I have found some words that were not separated and  two instances of a too informal language. Overall, the use of English is very good.

Response: We have now improved the expression of certain sentences and phrases as suggested by the reviewer.

  1. Martinez-Forero, I., Rouzaut, A., Palazon, A., Dubrot, J., & Melero, I. (2009). Lysine 63 polyubiquitination in immunotherapy and in cancer-promoting inflammation. Clinical cancer research : an official journal of the American Association for Cancer Research, 15(22), 6751–6757.
  2. Ma, Q., Ruan, H., Peng, L., Zhang, M., Gack, M. U., & Yao, W. D. (2017). Proteasome-independent polyubiquitin linkage regulates synapse scaffolding, efficacy, and plasticity. Proceedings of the National Academy of Sciences of the United States of America, 114(41), E8760–E8769.

Round 2

Reviewer 2 Report

Comments and Suggestions for Authors

The authors responded to all of my points. I appreciate the work put into improving the manuscript and the close attention paid to details. 

There are only some small changes that are needed in order for the manuscript to be ready for publication:

1) line 196-197: "based on which we will investigate whether miR-19a/b is involved in the process of bone invasion in PA" please refer only to miR-19a ( since this is the miRNA that was investigated) and delete future tense "will": "based on which we decided to investigate whether miR-19a (part of miR19a/b) is involved in the process of bone invasion in PA"

2) lines 590-592: " Studies  have illustrated the role of miR-19 in promoting osteoclast differentiation[21, 22], while some have demonstrated that miR-19 promotes osteoblast activation[44-46]. " - please specify whether you are referring to miR-19a, miR-19b or miR-19a/b

3) Move the initial part of the text for Figure 1 below the picture

4) I did not find the densitometry graph for TRAF6/GAPDH that was included in response to reviewer, but not in the manuscript (please add it to the manuscript). Also, include in Figure 3 the densitometry results for Ub/GAPDH to prove the change in Ub level 

5) include these references for bone mass/metastasis driven by miR-19a/b in the discussion section

  https://www.nature.com/articles/s41467-021-25473-y

https://www.spandidos-publications.com/10.3892/or.2017.6096

https://oa-fund.ub.uni-muenchen.de/id/eprint/403/1/EMBO%20Mol%20Med%20-%202022%20-%20Taipaleenm%20ki%20-%20Antagonizing%20microRNA%E2%80%9019a%20b%20augments%20PTH%20anabolic%20action%20and%20restores%20bone%20mass%20in.pdf 

6) Correct section 2.2. - "animal studies"  with "Animal studies", correct : "Figure2" with "Figure 2"

7) line 625, conclusion: "What’s more" - remove and correct , this is an informal wording

Comments on the Quality of English Language

The quality of English is very good. 

Author Response

Dear Editors and Reviewers:

Thank you for your comments concerning our manuscript entitled “miR-19a/CYLD axis regulates pituitary adenoma bone invasion by promoting osteoclast differentiation” (cancers-2765320). These comments are all valuable and very helpful for revising and improving our paper, as well as the important guiding significance to our researches. We have studied comments carefully and have made correction which we hope meet with approval.

We will respond to comments as follows:

1) line 196-197: "based on which we will investigate whether miR-19a/b is involved in the process of bone invasion in PA" please refer only to miR-19a ( since this is the miRNA that was investigated) and delete future tense "will": "based on which we decided to investigate whether miR-19a (part of miR19a/b) is involved in the process of bone invasion in PA"

Response: We have revised the statements according to the comment.

2) lines 590-592: " Studies  have illustrated the role of miR-19 in promoting osteoclast differentiation[21, 22], while some have demonstrated that miR-19 promotes osteoblast activation[44-46]. " - please specify whether you are referring to miR-19a, miR-19b or miR-19a/b

Response: We will use "miR19a/b" as the term, because the articles we cite contain the roles both of miR-19a and miR-19b for description.

3) Move the initial part of the text for Figure 1 below the picture

Response: We have adjusted the position of Figure 1 so that the figure legend of Figure 1 is below the figure.

4) I did not find the densitometry graph for TRAF6/GAPDH that was included in response to reviewer, but not in the manuscript (please add it to the manuscript). Also, include in Figure 3 the densitometry results for Ub/GAPDH to prove the change in Ub level 

Response: We have added the statistical graphs of Co-IP and WB in Figure 3G.

5) include these references for bone mass/metastasis driven by miR-19a/b in the discussion section

  https://www.nature.com/articles/s41467-021-25473-y

https://www.spandidos-publications.com/10.3892/or.2017.6096

https://oa-fund.ub.uni-muenchen.de/id/eprint/403/1/EMBO%20Mol%20Med%20-%202022%20-%20Taipaleenm%20ki%20-%20Antagonizing%20microRNA%E2%80%9019a%20b%20augments%20PTH%20anabolic%20action%20and%20restores%20bone%20mass%20in.pdf 

Response: We added these relevant citations and reorganised the statement in the discussion section

6) Correct section 2.2. - "animal studies"  with "Animal studies", correct : "Figure2" with "Figure 2"

Response: We have revised this according to the comment.

7) line 625, conclusion: "What’s more" - remove and correct , this is an informal wording

Response: We have deleted the phrase "What’s more" and reorganised the statement.

We sincerely hope that editors and reviewers could approve our revision of the article.